# Postoperative Pain Management after Full Endoscopic Lumbar Discectomy: An Observational Study

**DOI:** 10.3390/medicina58121817

**Published:** 2022-12-09

**Authors:** Guang-Xun Lin, Li-Wei Sun, Shang-Wun Jhang, Chien-Min Chen, Gang Rui, Bao-Shan Hu

**Affiliations:** 1Department of Orthopedics, The First Affiliated Hospital of Xiamen University, School of Medicine, Xiamen University, Xiamen 361000, China; 2The School of Clinical Medicine, Fujian Medical University, Fuzhou 350122, China; 3Division of Neurosurgery, Department of Surgery, Changhua Christian Hospital, Changhua 500209, Taiwan; 4Department of Leisure Industry Management, National Chin-Yi University of Technology, Taichung 41170, Taiwan; 5School of Medicine, Kaohsiung Medical University, Kaohsiung 80708, Taiwan

**Keywords:** lumbar disc herniation, full endoscopic lumbar discectomy, rebound pain, postoperative pain, FELD

## Abstract

*Background*: Surgical incision pain, rebound pain, and recurrence can manifest themselves in different forms of postoperative pain after full endoscopic lumbar discectomy (FELD). This study aims to evaluate various postoperative pains after FELD and summarize their characteristics. *Methods*: Data about the demographic characteristics of patients, pain intensity, and functional assessment results were collected from January 2016 to September 2019. Clinical outcomes including Oswestry Disability Index (ODI) and visual analog scale (VAS) scores, were obtained. *Results*: A total of 206 patients were enrolled. ODI and VAS of the patients significantly decreased after FELD at 12-month follow-up. A total of 193 (93.7%) patients had mild surgical incision pain after FELD and generally a VAS < 4, and it mostly resolved on its own within 3 days. A total of 12 (5.8%) patients experienced rebound pain, which was typically characterized by pain (mainly leg pain with or without back pain), generally occurring within 2 weeks after FELD and lasting < 3 weeks. The pain levels of rebound pain were equal to or less than those of preoperative pain, and generally scored a VAS of < 6. The recurrence rate was 4.4%. Recurrence often occurs within three months after surgery, with the pain level of the recurrence being greater than or equal to the preoperative pain. *Conclusions*: Different types of postoperative pain have their own unique characteristics and durations, and treatment options are also distinct. Conservative treatment and analgesia may be indicated for rebound pain and surgical incision pain, but recurrence usually requires surgical treatment.

## 1. Introduction

Full endoscopic lumbar discectomy (FELD) has evolved rapidly in the treatment of degenerative lumbar spine disease [1,2,3,4]. Compared with traditional open surgery, FELD has evident advantages. These include a low volume of blood loss, less trauma, faster recovery, and a shorter length of hospitalization [5,6]. However, some patients still experience pain symptoms for a period of time after FELD [7,8,9]. The manifestations of postoperative pain after FELD are diverse, and the characteristics of pain have different causes [10]. Surgical incision pain, inadequate decompression, epidural hematoma, nerve root injury, and recurrence can manifest themselves in different forms of postoperative pain. In addition, during our clinical experience, we observed that some patients experience pain relief immediately after surgery, but after a few days they may feel mild pain in the back or leg, soreness and weakness in the back, or leg numbness, especially when they get up and stand or walk, but it is usually not too serious. This condition was referred to as rebound pain, and the pathological mechanism of recurrent transient pain after FELD is not yet fully elucidated. As a distinct pain phenomenon, it is not a notion that has gained widespread support. These postoperative pains vary in terms of presentation, appearance, duration, and degree. Some of these postoperative pains are temporary, while others require medication or surgical intervention. Obviously, if the surgeons have a good grasp of these postoperative pain characteristics, it will help them better address the patient’s pain and respond more accurately to the issue. 

Thus far, there is no specific research about transient rebound pain. Hence, some studies have retrospectively analyzed the recurrence of transient pain in some patients who underwent FELD. On this basis, the current study aimed to assess various postoperative pains that appears after FELD, and summarizes the characteristics of various postoperative pains. We also sought evaluate the incidence, clinical features, and long-term clinical effects of rebound pain after FELD, to assess the risk factors of rebound pain, and to provide appropriate clinical diagnosis and treatment.

## 2. Materials and Methods

This is a retrospective study. The current study was approved by the institutional review board of Changhua Christian Hospital (CCH No.190905). After a cautious review of the data, we retrospectively selected 328 consecutive patients who underwent FELD from January 2016 to September 2019. The inclusion criteria were as follows: (1) patients with disc herniation with unilateral radiculopathy, with/without mild stenosis (2) those with single-level symptoms, and (3) those who received failed conservative treatment for more than 3 months. Patients with spinal instability, multilevel symptoms, spondylolisthesis or scoliosis, moderate/severe spinal stenosis, infection, trauma, neoplasm, recurrence, or a history of prior spinal surgery at the index level were excluded from the research. 

In addition, based on the presence of recurrent transient pain after FELD, the patients were divided into rebound and non-rebound pain groups.

### 2.1. Surgical Technique 

Using the transforaminal approach, the patients received epidural anesthesia while in the prone position under fluoroscopic guidance to confirm the surgical level. The working channel was inserted over the dilator and along the guide wire after a skin incision was made. An endoscope (SPINENDOS GmbH, München, Germany) was inserted into the working channel. In some cases, foraminoplasty was required to enlarge the working space via the removal of some bony tissues using a reamer, trephine, or high-speed burr. Then, the disc fragments were removed, and an annuloplasty was performed.

### 2.2. Postoperative Management

Postoperative dehydration and neurotrophic drugs were routinely administered. Patients were monitored for 24 h postoperatively, and they moved freely on the bed. One day after FELD, patients with a waist brace performed back muscle exercises. Simultaneously, we instructed our patients to minimize strenuous activity (strenuous activity refers to those activities that are vigorous, high volume, high frequency, and confrontational, with excessive load on the cardiorespiratory function; under normal circumstances a heart rate of more than 120 beats per minute after exercise is called strenuous exercise. Most anaerobic exercises are strenuous exercises, such as running, soccer, basketball, and high-volume equipment fitness) for 3–4 weeks after surgery and to wear a lumbar brace for at least 4 weeks after surgery. 

### 2.3. Outcome Evaluation

Demographic data, including age, sex, smoking, body mass index (BMI), diagnosis, and surgical level, were collected. In addition, information regarding the operative time and complications was obtained.

Data about patient-reported functional outcomes, including visual analog scale (VAS) scores for back and leg pain and the Oswestry Disability Index (ODI) at each follow-up time, were prospectively collected. Moreover, VAS scores for surgical incision pain were monitored daily until 3 days postoperatively. The rebound pain group underwent weekly monitoring until 1 month and at 3, 6, and 12 (last f/u) months after surgery. The satisfaction rate was assessed based on the modified Macnab criteria (excellent or good outcomes) [11]. 

After discharge, we regularly conducted telephone follow-ups with patients to collect data on outcomes, including characteristics of pain, the occurrence of rebound pain, functional changes, and treatment efficacy. All of the data were collected by one research assistant.

Preoperative magnetic resonance imaging (MRI) was performed to assess for Modic change and Schmorl’s node. The herniated disc size and location were evaluated using the Michigan State University classification (Figure 1) [12]. Lumbar disc degeneration was evaluated according to the Pfirrmann grading system [13]. MRI was again performed 3 months after surgery to observe for residual mass or recurrence. All radiological data were evaluated by two senior spine surgeons who were blinded to the study.

### 2.4. Statistical Analysis

Quantitative variables were expressed as mean ± standard deviation. The two-sample *t*-test or the Kolmogorov–Smirnov test as well as the x^2^ test or the Fisher’s exact test were used to analyze continuous and categorical variables, respectively. The logistic regression model was used to assess the risk factors for rebound pain. Statistical analyses were performed using the Statistical Package for the Social Sciences software, version 24.0 (IBM Corp., Armonk, NY, USA). A *p*-value of <0.05 was considered statistically significant.

## 3. Results

### 3.1. General Information

122 patients were lost to follow-up. Finally, a total of 206 patients, including 99 men and 107 women, were enrolled in this study. The mean age and BMI of the participants were 36.7 ± 7.6 (range: 20–57) years and 22.1 ± 2.6 (range: 17.8–30.6) kg/m^2^, respectively. Moreover, there were 37 (17.9%) smokers, including 12 with lesions at the L3/4 level, 120 at the L4/5 level, and 74 at the L5/S1 level, presented with single-level lumbar disc herniation without stenosis. Only 11 patients exhibited Modic change at the index level, and 8 (3.9%) presented with Schmorl’s node at the index level. According to the Pfirrmann grading system, 63 presented with a grade I lesion, 130 with a grade II lesion, and 13 with a grade III lesion. The average operative time was 40.4 ± 7.0 (range: 25–54) mins.

### 3.2. Clinical Outcomes

In the current study, 193 out of 206 patients (93.7%) had mild surgical incision pain after FELD, and most of them could relieve the pain by themselves with appropriate analgesic drugs. The mean VAS for surgical incision pain decreased from 3.02 ± 0.68 on postoperative day-1 to 1.48 ± 0.59 on postoperative day-2 and 0.61 ± 0.69 on postoperative day-3. It seems that the surgical incision pain lasts about 3 days after FELD. In addition, most patients have VAS scores of less than 3, and a few patients reach 4.

One year after surgery, the VAS scores for back and leg pain significantly decreased from 5.6 ± 1.8 to 1.3 ± 1.1 and from 6.9 ± 1.5 to 1.2 ± 1.1, respectively (*p* < 0.05). Further, the ODI scores significantly improved from 49.3 ± 11.5 to 13.8 ± 5.5 (*p* < 0.05). 

Nine patients experienced recurrence, with an incidence rate of 4.4%. According to our study, recurrence often occurred within 3 months after surgery. In most cases, the pain level of recurrence is greater than or equal to the preoperative pain (Table 1). Among them, eight required an additional endoscopic lumbar discectomy, and one underwent open fusion surgery. 

### 3.3. Rebound Pain Group

In total, 12 (5.8%) patients experienced rebound pain. Table 2 and Table 3 depict the demographic data and clinical characteristics of patients (*n* = 12) with rebound pain. There were three men and nine women, with an average age of 38.4 ± 2.4 (range: 26–46) years. Patients commonly presented with single-level lumbar disc herniation without stenosis (*n* = 11 [91.7%]), and these included five patients with lesions at the L4/5 and seven at the L5/S1. Rebound pain often occurred within 2 weeks after surgery, and it lasted less than 3 weeks. In most cases, the pain level of rebound pain was less than or equal to the preoperative pain, and a high proportion of patients had VAS scores of <6 (Table 4). According to the modified Macnab criteria, eight and four patients presented with excellent and good outcomes, respectively.

### 3.4. Subgroup Analysis

There was no significant difference in terms of age, sex, smoking, BMI, diagnosis, surgical level, operative time, and preoperative clinical outcomes between the non-rebound and rebound pain groups (*p* > 0.05) (Table 4). Table 5 presents the baseline data on disc herniation between the two groups. The herniated disc was commonly located on the 2-B side, followed by the 2-AB side. Results showed no significant differences between the two groups in terms of disc herniation size and location. Changes in pain and functional recovery between the two groups were compared, and results showed no significant differences in early postoperative and short-term efficacy (Figure 2, Figure 3 and Figure 4). In the logistic regression analysis, age, sex, BMI, diagnosis, surgical level and approach, Modic change, disc degeneration, operative time, and the preoperative VAS score and ODI did not significantly affect the development of postoperative rebound pain (*p* > 0.05) (Table 6).

## 4. Discussion

At present, with the development of FELD and the accumulation of cases, increasing attention has been directed toward the treatment and prevention of complications. Previous studies have shown that this technique is associated with some complications, including nerve root injury, residual remnants, prominent recurrence, and postoperative transient/persistent pain [14,15]. Postoperative pain after FELD has various manifestations, and the different characteristics are correlated with varying causes. Further, the outcome and prognosis of clinical symptoms differ [16,17]. This study focused on various postoperative pains that occur after FELD and summarized the characteristics of various postoperative pains.

It is common to experience pain at the site of the incision. As the wound heals, the pain will decrease. Most of the pain and soreness will disappear when the wound is completely healed and the sutures or staples are removed. In open surgery, deep tissue soreness and pain usually last for one to two weeks. Similar to our study, most patients (93.7%) had mild surgical incision pain after surgery. However, it mostly resolved on its own within 3 days, which is a benefit of FELD’s lower trauma.

Previously, some experienced spine surgeons considered rebound pain only an empirical finding, and it was even disregarded by several surgeons. To date, there is no relevant study about this unique condition, and standardized diagnostic criteria are not available. The pathogenesis of rebound pain has not been fully elucidated, and it may be correlated with several factors. Although the nerve roots have good activity after adhesion release and decompression via FELD, they remain in an inflammatory edematous state due to surgical trauma and self-repair. Moreover, insufficient blood supply in the local vasculature can further aggravate inflammatory edema [18]. In our opinion, the removal of herniated disc tissues will cause the cavity to be filled with blood clots, and inflammatory edema can occur. Organization and absorption may take time, during which symptoms may aggravate. Furthermore, articular surface and capsule damage during FELD, resulting in facet joint degeneration and instability, may lead to the aggravation of postoperative symptoms. 

In the current study, the incidence rate of transient rebound pain is 5.8%. This is not lower than the incidence of various postoperative complications after FELD. Interestingly, rebound pain mostly occurs in the early postoperative period, and the pain is not too severe. We did not identify the precipitating factors of rebound pain based on the survey indexes of age, sex, BMI, diagnosis, surgical level, Modic change, disc degeneration, operative time, and preoperative VAS score and ODI in the logistic regression analysis. Regarding clinical efficacy, our study compared pain intensity and functional assessment results at least during the 1-year follow-up between the rebound and non-rebound pain groups. There were no significant differences between the two groups. Thus, postoperative rebound pain is a short-term symptom, and it has no significant effect on the final efficacy of FELD.

According to previous studies, the prevalence of recurrence after FELD can reach 2.8–11% [11,19,20]. In our study, the recurrence rate was 4.4%, which was within an acceptable range. As for the characteristics of pain, early recurrence pain symptoms are similar to transient rebound pain, and both have a transient pain remission period [21]. The difference is that the patient feels well after FELD and then suddenly suffers the same or more severe pain and numbness as before; this is not postoperative rebound pain but may be a recurrence.

Based on the findings of this study, the characteristics of postoperative pain are summarized as follows: (1) Most patients have mild surgical incision pain after FELD, and it mostly resolves on its own within 3 days, and VAS scores are generally lower than 4. Most of them can relieve pain on their own, and appropriate analgesics can be given when necessary. (2) Postoperative rebound pain has a distinct pathological process of regaining pain after relief, and its pain level is less than or equal to the preoperative pain (a high proportion of patients had VAS scores of <6). Rebound pain is typically characterized by pain (mainly leg pain with or without back pain) and usually occurs within 2 weeks after surgery and lasts <3 weeks. Rebound pain does not require surgical intervention; most of it can be relieved by adequate bed rest, and if necessary, analgesics and physical therapy are used to relieve the pain. Also, rebound pain did not affect long-term surgical outcomes. (3) Recurrence often occurs within 3 months after surgery, and the pain level is greater than or equal to the preoperative pain. Most patients require surgical intervention for recurrence.

In addition, several other complications of postoperative pain, such as residual remnants, hematomas, and nerve root injuries, should also be noted [22,23]. Special caution, particularly for the management of residual remnants and nerve root damage, should be taken if there is no pain during the remission period after surgery. If a residual mass is suspected, an MRI should be performed. In the current study, patients with rebound pain underwent MRI to exclude organic causes such as residual remnants and recurrent disc herniation. In a recent study, 38 (16.9%) patients presented with residual remnants after FELD based on postoperative MRI findings. Among them, three (1.3%) were symptomatic [24]. Therefore, although symptomatic residuals can result in reoperation, the watchful and wait strategy may be a suitable option for patients with asymptomatic residual remnants. Postoperative pain can also manifest as transient postoperative pain when a postoperative hematoma compresses a nerve. Drainage is usually not placed after FELD, and this may lead to the formation of hematomas, which takes a certain period of time. Moreover, it indirectly manifests as postoperative pain during the remission period. Because blood is a fluid, compression may not be limited to the nerve root involved before surgery [25]. If a hematoma develops after surgery, postoperative pain is distributed in multiple areas and is even involved in the development of cauda equina syndrome. In cases of transient rebound pain, multiple nerve root involvement and cauda equina syndrome are not observed. In an iatrogenic nerve injury, there is no pain remission period. This type of complication can commonly be observed during surgery, and pain caused by a nerve injury can develop immediately after surgery without a transient improvement process. Further, symptoms do not significantly improve. 

The current study had several limitations. That is, it was retrospective in nature, had a small sample size, and a short-term follow-up duration. Patient data that was lost during the follow-up may affect the results. In addition, there were no power calculations for our samples. Moreover, the definition of rebound pain after FELD is uncertain due to its diverse features, unclear etiology, and limited cases. Therefore, large multicenter prospective studies must be conducted in the future.

## 5. Conclusions

Different types of postoperative pain have their own unique characteristics and durations. Most patients gave feedback on the pain of the surgical incision, which could be relieved by themselves or by using analgesics appropriately to relieve the pain, and its duration, usually about 3 days. Approximately 5.8% of patients experienced rebound pain, and its level is less than or equal to the preoperative pain, as well as being mitigated by conservative treatment, sufficient bed rest, and analgesics. Rebound pain does not affect the final efficacy of FELD. Recurrence often occurs within three months after FELD, and the pain level is greater than or equal to the preoperative pain. In addition, most cases required surgical intervention. Careful differentiation between the various types of postoperative pain is needed to contribute to better patient relief.

## Figures and Tables

**Figure 1 medicina-58-01817-f001:**
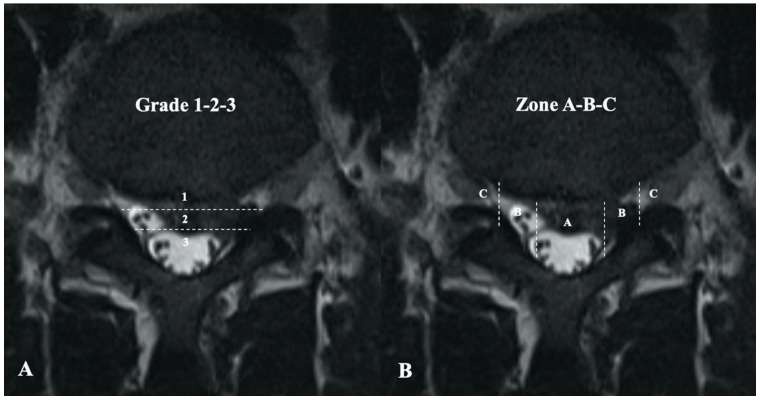
Michigan State University classification of herniated lumbar discs on magnetic resonance imaging. (**A**) Grading of disc herniation according to size. Grade 1 lesions have minimal impact and grade 3 have the most significant effect on nerve compression. (**B**) Zoning of the disc to identify location. Lesions in tighter zones (B and C) had more impact.

**Figure 2 medicina-58-01817-f002:**
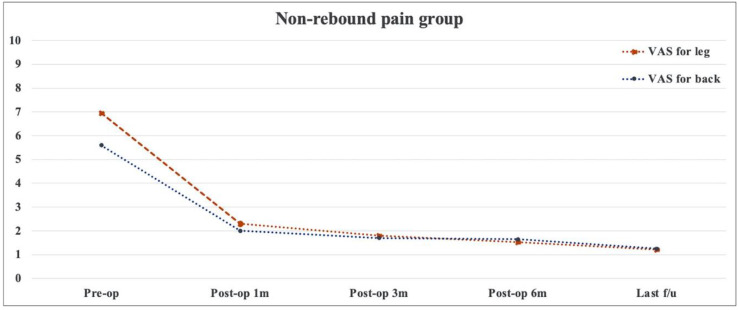
Changes in visual analog scale scores with time in the non-rebound pain group.

**Figure 3 medicina-58-01817-f003:**
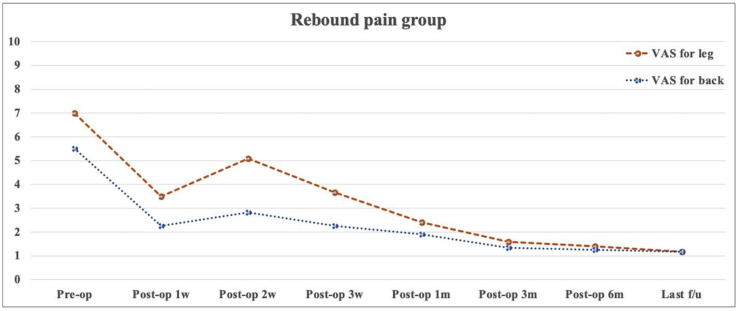
Changes in visual analog scale scores with time in the rebound pain group.

**Figure 4 medicina-58-01817-f004:**
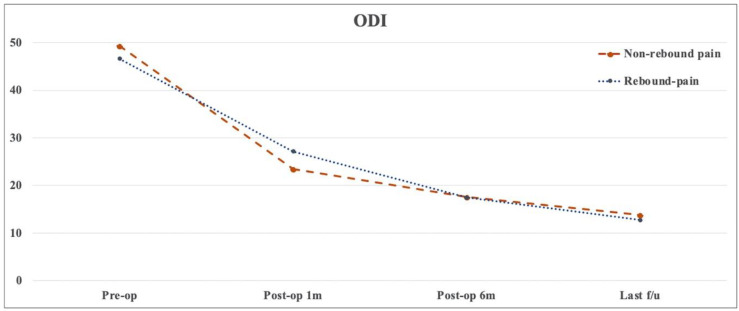
Changes in the Oswestry disability index scores with time for the non-rebound and rebound pain groups.

**Table 1 medicina-58-01817-t001:** Changes in VAS and ODI scores of recurrent pain in patients with time.

No.		1	2	3	4	5	6	7	8	9
Scores										
VAS (back)	1st pre-op	5	6	5	4	6	4	5	4	7
	1st post-op 1 w	2	2	2	1	2	3	2	2	3
	Recurrence time after 1st operation	3 weeks	10 weeks	6 weeks	8 weeks	2 weeks	7 weeks	6 weeks	4 weeks	5 weeks
	2nd pre-op	6	5	5	3	6	5	5	6	7
	2nd post-op 1 w	3	2	3	2	3	2	3	3	4
	2nd post-op 1 mo	2	2	2	1	2	1	2	2	3
	2nd post-op 6 mo	1	0	1	1	0	0	0	0	3
	Last f/u	1	0	1	1	0	0	0	0	2
VAS (leg)	1st pre-op	7	7	8	7	7	6	9	7	8
	1st post-op 1 w	3	4	3	2	3	2	3	2	3
	2nd pre-op	8	7	7	8	9	8	10	7	9
	2nd post-op 1 w	4	3	3	4	4	3	5	3	3
	2nd post-op 1 mo	4	3	3	3	3	3	4	4	3
	2nd post-op 6 mo	3	1	2	2	0	2	2	2	1
	Last f/u	2	1	2	1	1	1	2	1	1
ODI	1st pre-op	43.33	38.0	51.5	47.75	52.0	46.25	43.33	37.7	53.5
	2nd pre-op	38.85	43.35	54.0	48.5	48.75	49.25	39.75	35.33	51.25
	2nd post-op 6 mo	12.0	14.0	18.0	13.33	18.0	24.0	24.0	18.0	22.0
	Last f/u	10.0	4.0	4.0	8.0	10.0	16.0	8.0	12.0	10.0

VAS: visual analogue scale; ODI: Oswestry disability index.

**Table 2 medicina-58-01817-t002:** Characteristics of patients in the rebound pain group.

No.	Sex/Age (y)	Diagnosis	Level	Occurrence Time	Duration	Macnab Criteria
1	F/26	LDH	L4-5	10 d	1 w	Good
2	F/29	LDH	L4-5	4 d	2 w	Excellent
3	F/30	LDH	L4-5	11 d	2 w	Excellent
4	F/35	LDH	L4-5	13 d	2 w	Excellent
5	M/35	LDH	L5-S1	5 d	1 w	Good
6	F/37	LDH	L4-5	8 d	1 w	Good
7	M/37	LDH	L5-S1	3 d	2 w	Excellent
8	F/41	LDH	L4-5	14 d	3 w	Excellent
9	F/41	LDH	L4-5	2 d	1 w	Excellent
10	M/42	LDH	L5-S1	10 d	2 w	Excellent
11	F/44	LDH	L5-S1	8 d	1 w	Excellent
12	F/46	LDH with stenosis	L5-S1	6 d	1 w	Good

**Table 3 medicina-58-01817-t003:** Changes in VAS and ODI scores of rebound pain in patients with time.

No.		1	2	3	4	5	6	7	8	9	10	11	12
Scores													
VAS (back)	Pre-op	3	4	6	8	3	7	8	4	5	5	5	8
	Post-op 1 w	2	2	2	2	2	3	3	2	3	2	1	3
	Post-op 2 w	3	3	3	3	2	3	3	4	3	2	2	3
	Post-op 3 w	2	2	2	2	2	3	3	3	2	2	2	2
	Post-op 1 mo	0	2	1	2	2	3	2	3	2	2	2	2
	Post-op 3 mo	0	1	1	2	1	2	2	2	1	1	2	1
	Post-op 6 mo	0	1	1	2	1	2	2	2	1	0	1	2
	Last f/u	0	0	1	2	1	2	2	2	1	0	1	2
VAS (leg)	Pre-op	6	7	7	7	8	7	9	7	6	6	7	7
	Post-op 1 w	2	2	2	2	2	6	3	2	3	5	7	6
	Post-op 2 w	5	6	6	5	2	5	7	6	5	4	6	4
	Post-op 3 w	2	4	5	5	4	3	5	5	5	2	2	2
	Post-op 1 mo	2	2	2	1	3	3	4	4	2	2	2	2
	Post-op 3 mo	2	1	2	1	1	2	2	2	1	1	2	2
	Post-op 6 mo	2	1	2	2	0	2	1	2	0	2	1	2
	Last f/u	2	0	0	1	1	1	2	1	1	2	1	2
ODI	Pre-op	31.1	42.2	30.0	52.0	50.0	40.0	50.0	37.7	60.0	59.4	39.0	68.4
	Post-op 1 mo	26.0	23.2	26.4	22.4	39.6	31.0	33.3	25.0	24.0	22.0	26.4	26.6
	Post-op 6 mo	18.0	12.0	18.0	11.1	18.0	22.2	24.0	12.0	16.0	12.0	22.0	24.0
	Last f/u	14.0	2.0	4.0	8.0	13.0	13.3	18	12.0	12.0	15.0	16.0	20.0

VAS: visual analogue scale; ODI: Oswestry disability index.

**Table 4 medicina-58-01817-t004:** Demographic data of the two groups.

Group	Non-Rebound Pain (*n* = 194)	Rebound-Pain (*n* = 12)	*p*-Value [95% CI]
Age (years)	36.6 ± 7.6	38.4 ± 8.4	0.41 [−6.33~2.61]
Sex			
Female	98	9	0.14
Male	96	3	
BMI (kg/m^2^)	21.9 ± 2.5	23.3 ± 3.6	0.08 [−2.85~0.16]
Smoking			
Yes	15	0	0.13
No	179	12	
Diagnosis			
LDH	179	11	1.0
LDH with stenosis	15	1	
Surgical level			
L3-4	12	0	0.65
L4-5	113	7	
L5-S1	69	5	
Modic change			
Yes	10	1	1.0
No	184	11	
Schmorl’s Nodes			
Yes	8	0	1.0
No	186	12	
Pfirrmann			
Grade II	59	4	0.92
Grade III	123	7	
Grade IV	12	1	
Pre-op VAS back	5.7 ± 1.8	5.2 ± 1.5	0.27 [−0.52~1.55]
Pre-op VAS leg	6.9 ± 1.5	7.0 ± 1.7	0.93 [−0.94~0.85]
Pre-op ODI	49.2 ± 11.6	51.6 ± 10.5	0.45 [−9.19~4.33]
Operative time (mins)	40.4 ± 6.9	39.9 ± 7.9	0.83 [−3.63~4.64]
Recurrence			
Yes	9	0	1.0
No	185	12	
Surgical incision pain			
Yes	182	11	<0.01
No	12	1	

VAS: visual analogue scale; ODI: Oswestry disability index.

**Table 5 medicina-58-01817-t005:** The size and location of the herniated disc in the two groups (based on MSU classification).

Group	Non-Rebound Pain (*n* = 194)	Rebound-Pain (*n* = 12)	*p*-Value
1-A	0	0	0.78
1-B	5	0	
1-C	3	0	
2-AB	40	5	
2-A	4	0	
2-B	119	6	
2-C	12	1	
3-AB	8	0	
3-A	0	0	
3-B	3	0	

**Table 6 medicina-58-01817-t006:** Logistic regression analyses for the risk factors of rebound pain.

Risk Factors	OR	*p*-Value
Age (years)		
20–39	Reference	0.70
40–57	1.41	
Sex		
Male	Reference	0.16
Female	0.35	
BMI (kg/m^2^)		
<24	Reference	0.09
≥24	0.28	
Diagnosis		
LDH	Reference	0.99
LDH with stenosis	0.99	
Surgical level		
L3–4	Reference	
L4–5	0	0.00
L5-S1	0.79	0.85
Modic change		
No	Reference	0.59
Yes	0.49	
Pfirrmann		
Grade II	Reference	
Grade III	0.64	0.73
Grade IV	0.55	0.63
Operative time (mins)		
<40	Reference	0.87
≥40	1.12	
Pre-op VAS (back)		
<4	Reference	
4–6	3.48	0.26
≥7	1.85	0.51
Pre-op VAS (leg)		
<4	Reference	
4–6	0	0.99
≥7	0.78	0.73
Pre-op ODI		
<40	Reference	
40–55	0.69	0.74
≥56	1.79	0.47

BMI: body mass index; VAS: visual analog scale; ODI: Oswestry disability index.

## Data Availability

The datasets are presented within the manuscript.

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
