# Peer review of "Postoperative Pain Management after Full Endoscopic Lumbar Discectomy: An Observational Study"

_medicina, 2022, doi:10.3390/medicina58121817_

Round 1
Reviewer 1 Report
Feedback:
Thank you for inviting me to review the manuscript ‘Postoperative pain management after full endoscopic lumbar discectomy: an observational study’.
Introduction
· Line36/37: Do you mean the development of FELD ‘has been’ rapid?
· Line 39: You could reference the paper https://link.springer.com/article/10.1007/s00586-019-06272-y, which clearly demonstrates rebound pain, but does not refer to it as that.
· Line 59: insert ‘a secondary purpose’ after ‘also’.
Methods
· Line 63: Please state what type of study this is [retrospective cohort?].
· Line 64: Which relevant guidelines and regulations?
· Line 66: How many people where operated on in total? How many did you exclude?
· Line 73: Please explain how you identified that they had rebound pain or not, and why you would know that?
· Line 86: Please explain what a waist / lumbar brace is and why it is indicated. Please define strenuous activity?
· Line 96: Why were only the rebound group monitored weekly until 1 month, then 3, 6 and 12 months? How did you know who experienced rebound pain – please explain it is not clear.
· Line 100: Was telephoning part of your standard service? It sounds prospective. It is confusing.
· Line 115: How do you express qualitative variables as numbers? This is unclear.
· Line 116: There are no power calculations for your sample. Please comment here and in the limitations
· Line 118: please state which variables are being tested in each test in order to meet your aims / purpose. Your aims do not mention prediction. The first aim is to assess various postoperative pains [What do mean by various postoperative pains? Do you mean leg and back or types of pain, e.g., nociceptive, neuropathic, or nociplastic?] and their characteristics. The second is to evaluate incidence, clinical features, and effects.
Results
· Line 124: why were 122 lost to follow up? Do you have their characteristics?
· Line 127: What are the 190 (92.2%) of patients referring to?
· Line 134: insert ‘the’ after ‘In’.
· Table 1: Why don’t you present your weekly pain data up to one month? Your reporting of pain scores for leg band back is not consistent – why?
· Table 4: please provide the 95% CI alongside the P value.
· Figure 4: the title is misleading. Please replace ‘between’ with ‘for’.
Discussion
· Line 192: please explain this statement “Postoperative pain after FELD has various manifestations, and the different characteristics are correlated with varying causes”.
· Lines 210-214 need citations.
· Lines 218: Do you think you included relevant independent variables to predict rebound pain?
· Line 226: I think the discussion about rebound pain and recurrence is important to manage patient expectations. Could you predict recurrence rather than rebound pain?
· Line 228: is this acceptable range set locally, nationally, evidence-based?
· Line 242: When is physical therapy indicated following bed rest? Is bed rest evidence based? There is literature to say the contrary.
· Line 257: please reference the statement about hematoma formation and pain.
· Line 275: How should you manage patient expectations in terms of rebound pain?
General: Whilst your observations around rebound pain are interesting and have clinical relevance, you miss the opportunity to benefit the patient in terms of managing postop expectations. Your terminology regarding pain is misleading and out of date. You lack references to the latest research.
Author Response
Reviewer 1:
Feedback:
Thank you for inviting me to review the manuscript ‘Postoperative pain management after full endoscopic lumbar discectomy: an observational study’.
Introduction
- Line36/37: Do you mean the development of FELD ‘has been’ rapid?
ïƒ Thank you for your question. We changed the sentence as “Full endoscopic lumbar discectomy (FELD) has evolved rapidly in the treatment of degenerative lumbar spine disease.”
- Line 39: You could reference the paper https://link.springer.com/article/10.1007/s00586-019-06272-y, which clearly demonstrates rebound pain, but does not refer to it as that.
ïƒ Thank you for your kind comments. According to your comments, we added the reference in manuscript.
- Line 59: insert ‘a secondary purpose’ after ‘also’.
ïƒ Thank you for your kind comments. According to your comments, we inserted ‘a secondary purpose’ after ‘also’.
Methods
- Line 63: Please state what type of study this is [retrospective cohort?].
ïƒ Thank you for your kind comments. This is a retrospective study.
- Line 64: Which relevant guidelines and regulations?
ïƒ Thank you for your kind comments. This sentence is redundant and we have removed.
- Line 66: How many people where operated on in total? How many did you exclude?
ïƒ Thank you for your kind comments. After a cautious review of data, we retrospectively selected 328 consecutive patients who underwent FELD from Jan 2016 to September 2019. 122 patients were lost to follow-up. Finally, a total 206 patients, including 99 men and 107 women, were enrolled in this study.
- Line 73: Please explain how you identified that they had rebound pain or not, and why you would know that?
ïƒ Thank you for your questions. During our clinical experience, we observed that some patients experience pain relief immediately after surgery, but after a few days they may feel mild pain in the back or leg, or with soreness and weakness in the back, or leg numbness, especially when they get up and stand or walk, but it is usually not too serious. This condition was referred to as rebound pain, and the pathological mechanism of recurrent transient pain after FELD is not yet fully elucidated. As a distinct pain phenomenon, it is not a notion that has gained widespread support.
- Line 86: Please explain what a waist / lumbar brace is and why it is indicated. Please define strenuous activity?
ïƒ Thank you for your questions. Patients with herniated discs may need to wear a lumbar girth during the recovery period to compensate for the pressure on the lumbar discs, relieve the accumulation of joint stress and maintain joint stability on the one hand, and reduce muscle tension and maintain normal blood circulation on the other.
Strenuous activities are those that are violent, with large movements, high frequency, and confrontational, and that put too much load on the cardiorespiratory function... Under normal circumstances, the heart rate after exercise exceeds 120 beats per minute, are called strenuous exercise. Most of the anaerobic exercises are strenuous exercises, such as running, soccer, basketball, high volume equipment fitness, etc.
- Line 96: Why were only the rebound group monitored weekly until 1 month, then 3, 6 and 12 months? How did you know who experienced rebound pain – please explain it is not clear.
ïƒ Thank you for your questions. We have noticed in our early clinical work that some patients develop transient rebound pain after surgery. We followed up each patient included in this study by telephone at 1 month postoperatively, during which some patients would revisit the clinic because of pain, and we will continue to follow up on the evolution of pain in these patients. The reason why the non-rebound pain group did not give a VAS and ODI score at 1 month postoperatively is that there is no transient pain fluctuation curve as in the case of rebound pain.
- Line 100: Was telephoning part of your standard service? It sounds prospective. It is confusing.
ïƒ Thank you for your questions. It's not the standard service. We hired it specifically for this study as a research secretary.
- Line 115: How do you express qualitative variables as numbers? This is unclear.
ïƒ Thank you for your questions. Quantitative variables were expressed in average ± standard deviation and qualitative variables in numbers and percentage.
- Line 116: There are no power calculations for your sample. Please comment here and in the limitations
ïƒ Thank you for your kind comments. We added in the limitation section.
- Line 118: please state which variables are being tested in each test in order to meet your aims / purpose. Your aims do not mention prediction. The first aim is to assess various postoperative pains [What do mean by various postoperative pains? Do you mean leg and back or types of pain, e.g., nociceptive, neuropathic, or oncoplastic?] and their characteristics. The second is to evaluate incidence, clinical features, and effects.
ïƒ Thank you for your questions. The first aim is to assess various postoperative pain that appears after FELD, and summarizes the characteristics of various postoperative pains. This pain includes the back or leg and can be nociceptive or neuropathic in nature.
The second was to assess the incidence of postoperative pain, especially rebound pain, clinical features including VAS/ODI, and the duration of surgery and postoperative complications.
Results
- Line 124: why were 122 lost to follow up? Do you have their characteristics?
ïƒ Thank you for your questions. These patients included 65 female patients and 57 male patients, with 5 surgical segments in L3-4, 84 at L4-5 and 33 at L5-S1.
- Line 127: What are the 190 (92.2%) of patients referring to?
ïƒ Thank you for your questions. There may have been some errors, we have removed this sentence.
- Line 134: insert ‘the’ after ‘In’.
ïƒ Thank you for your kind comments. We inserted.
- Table 1: Why don’t you present your weekly pain data up to one month? Your reporting of pain scores for leg band back is not consistent – why?
ïƒ Thank you for your questions. It so happens that these recurrent patients did not have a transient rebound pain after the second surgery, so no weekly change in pain index is given.
- Table 4: please provide the 95% CI alongside the P value.
ïƒ Thank you for your kind comments. We added.
- Figure 4: the title is misleading. Please replace ‘between’ with ‘for’.
ïƒ Thank you for your kind comments. We changed.
Discussion
- Line 192: please explain this statement “Postoperative pain after FELD has various manifestations, and the different characteristics are correlated with varying causes”.
ïƒ Thank you for your questions. Based on the findings of this study, the characteristics of postoperative pains are summarized as follows: 1) most patients have mild surgical incision pain after FELD, and it mostly resolved on its own within 3 days, as well as VAS scores are generally lower than 4. Most of them can relieve pain on their own, and appropriate analgesics can be given when necessary. 2) postoperative rebound pain has a distinct pathological process of regaining pain after relief and its pain level is less than or equal to the preoperative pain (a high proportion of patients had VAS scores of < 6). Rebound pain, typically characterized by pain (mainly leg pain with or without back pain) and usually occurs within 2 weeks after surgery and lasts < 3 weeks. Rebound pain does not require surgical intervention, most of which can be relieved by adequate bed rest, and if necessary, analgesics and physical therapy are used to relieve the pain. Also, rebound pain did not affect long-term surgical outcomes. 3) recurrence often occurred within 3 months after surgery, and the pain level is greater than or equal to the preoperative pain. Most patients require surgical intervention when recurrence.
- Lines 210-214 need citations.
ïƒ Thank you for your kind comments. The pathogenesis of rebound pain is not fully elucidated, and it may be correlated with several factors. This is what we had in mind, and in order not to cause misunderstanding, we added "in our opinion" in front of it.
- Lines 218: Do you think you included relevant independent variables to predict rebound pain?
ïƒ Thank you for your kind comments. In the logistic regression analysis, we did not identify the provocative factors of rebound pain based on the survey indexes in terms of age, sex, BMI, diagnosis, surgical level, Modic change, disc degeneration, operative time, and preoperative VAS score and ODI.
- Line 226: I think the discussion about rebound pain and recurrence is important to manage patient expectations. Could you predict recurrence rather than rebound pain?
ïƒ Thank you for your questions. Based on the findings of this study, the characteristics of postoperative pains are summarized as follows: 1) most patients have mild surgical incision pain after FELD, and it mostly resolved on its own within 3 days, as well as VAS scores are generally lower than 4. Most of them can relieve pain on their own, and appropriate analgesics can be given when necessary. 2) postoperative rebound pain has a distinct pathological process of regaining pain after relief and its pain level is less than or equal to the preoperative pain (a high proportion of patients had VAS scores of < 6). Rebound pain, typically characterized by pain (mainly leg pain with or without back pain) and usually occurs within 2 weeks after surgery and lasts < 3 weeks. Rebound pain does not require surgical intervention, most of which can be relieved by adequate bed rest, and if necessary, analgesics and physical therapy are used to relieve the pain. Also, rebound pain did not affect long-term surgical outcomes. 3) recurrence often occurred within 3 months after surgery, and the pain level is greater than or equal to the preoperative pain. Most patients require surgical intervention when recurrence.
- Line 228: is this acceptable range set locally, nationally, evidence-based?
ïƒ Thank you for your questions. These data are based on reference reports.
- Line 242: When is physical therapy indicated following bed rest? Is bed rest evidence based? There is literature to say the contrary.
ïƒ Thank you for your questions. This is our objective experience concluded from the follow-up of 206 patients included in this study.
- Line 257: please reference the statement about hematoma formation and pain.
ïƒ Thank you for your questions. It should be “Postoperative pain can also manifest as a transient postoperative pain when a postoperative hematoma compresses a nerve.”
- Line 275: How should you manage patient expectations in terms of rebound pain?
ïƒ Thank you for your questions. Rebound pain, typically characterized by pain (mainly leg pain with or without back pain) and usually occurs within 2 weeks after surgery and lasts < 3 weeks. Rebound pain does not require surgical intervention, most of which can be relieved by adequate bed rest, and if necessary, analgesics and physical therapy are used to relieve the pain. Also, rebound pain did not affect long-term surgical outcomes.
General: Whilst your observations around rebound pain are interesting and have clinical relevance, you miss the opportunity to benefit the patient in terms of managing postop expectations. Your terminology regarding pain is misleading and out of date. You lack references to the latest research.
ïƒ Thank you for your kind comments. The manifestations of postoperative pain after FELD are diversified, and the characteristics of pain have different causes. Surgical incision pain, inadequate decompression, epidural hematoma, nerve root injury, and recurrence can manifest themselves in different forms of postoperative pain. In addition, during our clinical experience, we observed that some patients experience pain relief immediately after surgery, but after a few days they may feel mild pain in the back or leg, or with soreness and weakness in the back, or leg numbness, especially when they get up and stand or walk, but it is usually not too serious. This condition was referred to as rebound pain, and the pathological mechanism of recurrent transient pain after FELD is not yet fully elucidated. As a distinct pain phenomenon, it is not a notion that has gained widespread support. These post-operative pains vary in terms of presentation, appearance, duration and degree. Some of these postoperative pains are temporary, while others require medication or surgical intervention. Obviously, if the surgeons have a good grasp of these postoperative pain characteristics, it will help to better address the patient's pain and respond more accurately to the issue.
Thus far, there is no specific research about transient rebound pain. Hence, some studies retrospectively analyzed the recurrence of transient pain in some patients who underwent FELD. On this basis, the current study aimed to assess various postoperative pain that appears after FELD, and summarizes the characteristics of various postoperative pains. Also, a secondary purpose to evaluate the incidence, clinical features, and long-term clinical effects of rebound pain after FELD and to provide appropriate clinical diagnosis and treatment.
Different natures of postoperative pain have their own unique pain characteristics and durations. Most patients gave feedback on the pain of the surgical incision, which could be relieved by themselves or by using analgesics appropriately to relieve the pain, and its duration usually about 3 days. Approximately 5.8% of patients experienced re-bound pain, and its pain level is less than or equal to the preoperative pain, as well as it could be mitigated by conservative treatment and sufficient bed rest and analgesics. Rebound pain does not affect the final efficacy of FELD. Recurrence often occurred within 3 months after FELD, and the pain level is greater than or equal to the preoperative pain, also most cases required surgical intervention. Careful differentiation between the various types of postoperative pain is needed to contribute to better patient relief.

Reviewer 2 Report
The authors examined postoperative pain outcomes after full endoscopic lumbar discectomy (FELD, transforaminal approach) as well as other factors such as recurrence rate, ODI, and VAS (leg/back). The overall data presented in this study are consistent with the current literature. The focus was on rebound pain after FELD, and a total of 206 FELD patients were included in the final retrospective data analysis, with 12 (5.8%) patients having rebound pain. However, no provoking factor was identified.
The article is well written and has a consistent and clear structure. As noted in the article, rebound pain after spine surgery is a controversial topic in the literature. It would have been questionable if the authors could identify the dominant factors contributing to rebound pain in the niche area like FELD. Although the authors do not provide any new scientific evidence, they give a clear and simple summary of the different postoperative pain in the subset of the FELD patient cohort (mild incisional pain, postoperative rebound pain, recurrent pain).
The problem of retrospective nature, small sample size, and short follow-up time was addressed. The 122 (37%) patients lost to follow-up out of the 328 initial cohort may greatly confound the results; therefore, a prospective study design is clearly needed. As stated in the postoperative management section "The use of a lumbar brace for at least 4 weeks after surgery", is not supported in medical evidence (1). In addition, it would be interesting to see if similar results are obtained in patients treated with FELD (interlaminar approach).
1) The efficacy of postoperative bracing after spine surgery for lumbar degenerative diseases: a systematic review, Davide Nasi, Mauro Dobran2, Giacomo Pavesi European Spine Journal
Author Response
Thank you for your kind comments. Different natures of postoperative pain have their own unique pain characteristics and durations. Most patients gave feedback on the pain of the surgical incision, which could be relieved by themselves or by using analgesics appropriately to relieve the pain, and its duration usually about 3 days. Approximately 5.8% of patients experienced rebound pain, and its pain level is less than or equal to the preoperative pain, as well as it could be mitigated by conservative treatment and sufficient bed rest and analgesics. Rebound pain does not affect the final efficacy of FELD. Recurrence often occurred within 3 months after FELD, and the pain level is greater than or equal to the preoperative pain, also most cases required surgical intervention. Careful differentiation between the various types of postoperative pain is needed to contribute to better patient relief.
Round 2
Reviewer 1 Report
Thank you for your work on the amendments. Whilst you have provided explanations in your response to the reviewer - these haven't always been reflected in changes to the manuscript.
Please see my further recommendations.
· Line 66: please insert ‘was’ after ‘purpose’
· Line 67: please insert an aim ‘to assess the risk factors of rebound pain”
· Line 92: please insert your definition of strenuous into the paper for the reader.
· Line 123: Qualitative variables are text and do not involve numbers. You have not included any qualitative data or analysis. Please remove the term qualitative.
Author Response
Thank you for your work on the amendments. Whilst you have provided explanations in your response to the reviewer - these haven't always been reflected in changes to the manuscript.
Please see my further recommendations.
- Line 66: please insert ‘was’ after ‘purpose’
ïƒ Thank you for your kind comments. We inserted.
- Line 67: please insert an aim ‘to assess the risk factors of rebound pain”
ïƒ Thank you for your kind comments. We inserted.
- Line 92: please insert your definition of strenuous into the paper for the reader.
ïƒ Thank you for your kind comments. We inserted.
- Line 123: Qualitative variables are text and do not involve numbers. You have not included any qualitative data or analysis. Please remove the term qualitative.
ïƒ Thank you for your kind comments. We removed this part.
